# Spatial approach for diagnosis of yield-limiting nutrients in smallholder agroecosystem landscape using population-based farm survey data

Stephen M. Ichami [1,2,3,4,5]*, George N. Karuku[1], Andrew M. Sila[2], Fredrick O. Ayuke[1], Keith D. Shepherd[2]

**1** Faculty of Agriculture, Department of Land Resource Management and Agricultural Technology, University of Nairobi, Kabete, Nairobi, Kenya, **2** World Agroforestry Centre (ICRAF), Gigiri, Nairobi, Kenya, **3** International Centre for Tropical Agriculture (CIAT-Kenya), Nairobi, Kenya, **4** Soil Biology Group, Wageningen University, Wageningen, The Netherland, **5** Soil Geography and Landscape Group, Wageningen University, Wageningen, The Netherlands

* stephen.ichami@gmail.com

**Data Availability Statement:** All relevant data are available on OSF: https://doi.org/10.17605/OSF.IO/EVWNT.

## Abstract

Adept use of fertilizers is critical if sustainable development goal two of zero hunger and agroecosystem resilience are to be achieved for African smallholder agroecosystems. These heterogeneous systems are characterized by poor soil health mainly attributed to soil nutrient depletion. However, conventional methods do not take into account spatial patterns across geographies within agroecosystems, which poses great challenges for targeted interventions of nutrient management. This study aimed to develop a novel population-based farm survey approach for diagnosing soil nutrient deficiencies. The approach embraces principles of land health surveillance of problem definition and rigorous sampling scheme. The advent of rapid soil testing techniques, like infrared spectroscopy, offers opportune avenues for high-density soil and plant characterization. A farm survey was conducted on 64 maize fields, to collect data on soil and plant tissue nutrient concentration and grain yield (GY) for maize crops, using hierarchical and purposive sampling. Correlations between soil test values with GY and biomass were established. The relationship between GY, soil NPK, and the tissue nutrient concentrations was evaluated to guide the setting up of localized critical soil test values. Diagnosis Recommendation Integrated System (DRIS) indices for total nitrogen (N), total phosphorus (P), and total potassium (K) were used to rank and map the prevalence of nutrient limitations. A positive correlation existed between plant tissue nutrient concentration with GY with $R^2$ values of 0.089, 0.033, and 0.001 for NPK, respectively. Soil test cut-off values were 0.01%, 12 mg kg$^{-1}$, 4.5 cmol$_c$ kg$^{-1}$ for NPK, respectively, which varied slightly from established soil critical values for soil nutrient diagnostics. N and K were the most limiting nutrients for maize production in 67% of sampled fields. The study demonstrates that a population-based farm survey of crop fields can be a useful tool in nutrient diagnostics and setting priorities for site-specific fertilizer recommendations. A larger-scale application of the approach is warranted.

**Funding:** This study was funded by Wageningen University as a sandwich PhD, CIAT-Kenya and World Agroforestry Centre (ICRAF) fellowship programs in collaboration with the University of Nairobi. We acknowledge initial comments on the concept of this paper by Prof. Ellis Hoffland and Dr. Jetse Stoorvogel of Wageningen University.The unlimited cooperation and support extended by International Plant Nutrition Institute (IPNI) through Dr. Shamie Zingore in carrying out the field research work in western Kenya are gratefully appreciated. We also acknowledge the support of the CGIAR research program on 'Water, Land and Ecosystems'.

**Competing interests:** The authors have declared that no competing interests exist.

## 1.0 Introduction

Smallholder agroecosystems support livelihoods of 1.2 billion people and are the backbone of the rural economy [1]. These agroecosystems play a significant role in food production, poverty alleviation and mitigation against hunger for rural populations [2]. The smallholder systems are characterized by soil fertility degradation [3], low quality germplasm [4], which constraint crop production [5,6]. Poor soil health, associated with nutrient limitation is a major consequence of low crop productivity and has resulted in declined yields for staple cereals such as maize [7,8]. Soil health is the capacity of soil to respond to agricultural intervention so that it continues to support both agricultural production and provision of other ecosystem services [9,10]. To mitigate the scourge of poor soil health in the smallholder agroecosystems, accurate and repeatable methods for determining nutrient deficiencies are a prerequisite for judicious fertilizer investments [7]. In addition, such investments require good agronomic practices, which enhance biological processes and maintain physical properties like soil structure that also influence soil health.

It is becoming apparent to policymakers and soil scientists that conventional methods used in nutrient diagnostic may not be efficient, due to soil heterogeneity, particularly nutrient variation. Studies provide evidence of heterogeneity within smallholder agroecosystem, which is among cardinal causes of nutrient use inefficiencies and poor crop response to fertilizer applications [11–14]. However, few studies propose ways of dealing with soil heterogeneity in relation to site-specific nutrient diagnostics, to inform decisions on fertilizer requirements [15]. Conventional approaches lack a *rigorous* framework, which can help farmers make *empirical* evidence-based decisions on nutrient management [7,10]. The conventional nutrient diagnostics can generally be summarized as follows.

i. Visual symptoms observation (histology), which entails inspection of deficiency symptoms of nutrients that are most limiting to crop growth [16,17]. The deficiency of individual nutrient produces characteristic effects on various organs of the plants [18]. For example, stunted growth and yellow-greenish color (chlorosis) on leaves are normally associated with N limitation [19]. Ability to recognize these particular effects forms the basis of the visual method, which is readily applied by smallholder farmers [18,20]. Farmers also use indigenous knowledge such as soil colour and the presence of a particular weed within their farm to diagnose whether a nutrient is limiting [20,21]. However, the symptoms observed could easily be misinterpreted with other plant stress (e.g., pest and diseases). Recent studies associate chlorosis with maize leaves necrosis disease [22], while chlorotic stirps have been attributed to the presence of maize streak virus [23]. Therefore, this requires histology to be complemented with other methods (e.g. soil or foliar testing–using rapid soil testing kits), to ascertain the actual cause of observed symptoms [21].

ii. Nutrient omission trials, where a single and/or few trials are established in a specified geographical location, to evaluate crop responses to fertilizer applications [24–26]. The diagnostic results obtained are limited to that specific locality, and when extrapolated to other regions, it may lead to incorrect diagnostic conclusions [27,28]. The assumption that few trials would represent soil heterogeneity, at the regional or landscape levels is rarely realistic. This limits the applicability of omission trials, to accurately diagnose spatial pattern of limiting nutrients, due to variation across trial sites, within the agroecosystem [12,29].

iii. Soil testing, may provide information on the limiting nutrients. Often, low soil test values signify a positive crop response to fertilizer application [30,31]. Soil test values need to be calibrated to crop response before they can be interpreted accurately [32], but the lack of data to calibrate soil tests is a major setback in many developing countries [33,34]. Soil

testing must be in tandem with plant tissue testing, which is a powerful tool for diagnosing micronutrient deficiencies that may prevent responses to macronutrients [35,36]. High costs of wet chemical analysis, restrict the applicability of soil testing for large area assessments of limiting nutrients [17,37].

Given the limitations of the conventional methods, alternative approaches for nutrient diagnosis for smallholder agroecosystems are necessary for the rationalization of fertilizer investments decisions. Population-Based Survey (PBS) for evaluation of disease prevalence has become popular in epidemiology because it is a rapid and reliable way to assess a patient's health condition within a given population [38,39]. This approach is particularly useful when monitoring disease patterns within human populations and designing targeted curative medical interventions [40,41]. There is potential for developing a nutrient diagnostic approach, a Population-based Farms Survey (PFS), which can be anchored on the principles of Land Health Surveillance (LHS) that are borrowed from PBS [7]. The LHS deploys a rigorous ground sampling scheme and uses proximal techniques (*e.g.* infrared spectroscopy) for cheaper and rapid nutrient diagnosis [7]. The approach was recently applied in smallholder landscapes of Malawi, where Beedy *et al.*, identified areas with high-risk land degradation and provided options for targeted management interventions [42]. Information on soil and plant nutrient relationships is collected, and statistical models are employed to provide population-based estimates of means, Diagnosis Recommendation Integrated System (DRIS) indices, and confidence intervals on nutrient limitations [7,10]. The developed DRIS indices can be used to diagnose and rank limiting nutrients [43]. This can be done in tandem with digital property mapping for the evaluation of spatial patterns and prevalence of limiting nutrients. Consequently, spatial variability patterns of nutritional constraints are identified at a landscape scale. This provides an evidence-based guide for nutrient management decisions within smallholder agroecosystems. The proposed approach has never been tested in its applicability in nutrient management for smallholder agroecosystems, particularly in heterogeneous maize farming systems. However, uncertainty in PFS may stem from lack of knowledge by smallholder farmers, wrong data interpretation, and variability in laboratory chemical analysis for the reference data, for developing spectral calibrations for predicting properties of soil and plant samples [44]. These limitations can be overcome by enhancing accuracy in data collection to obtain correct interpretation, and employing standard laboratory analytical methods such as infrared spectroscopy.

The overall objective of the study was to test the PFS approach for the nutrient diagnosis in the smallholder agroecosystems of western Kenya. The region was deemed as a suitable testing site because it is typified with heterogeneous parcels of smallholder maize fields. The study specifically aimed to: (i) evaluate NPK nutrients limitations using farm surveys soil and plant data, and (ii) use indices and map the spatial distribution of the nutrient deficiencies across the landscape. NPK are considered as the major nutrients limiting plant growth [45]. A hypothesis that the spatial pattern occurrence of NPK nutrient limitation is random within smallholder agroecosystems was tested. Previous studies conducted in western Kenya, have characterized the region with poor soil health [46].

## 2.0 Material and methods

### 2.1 Study area

The domain of interest is smallholder maize farms, situated within administrative sub-counties of Boro, Butere, Yala, Khwisero, and Ugunja and lies between 0°26′ - 0°18′ northern latitude; 33°58′ - 34°33′ eastern longitude (Fig 1), within the sub-humid Lower Midland zone [43]. The

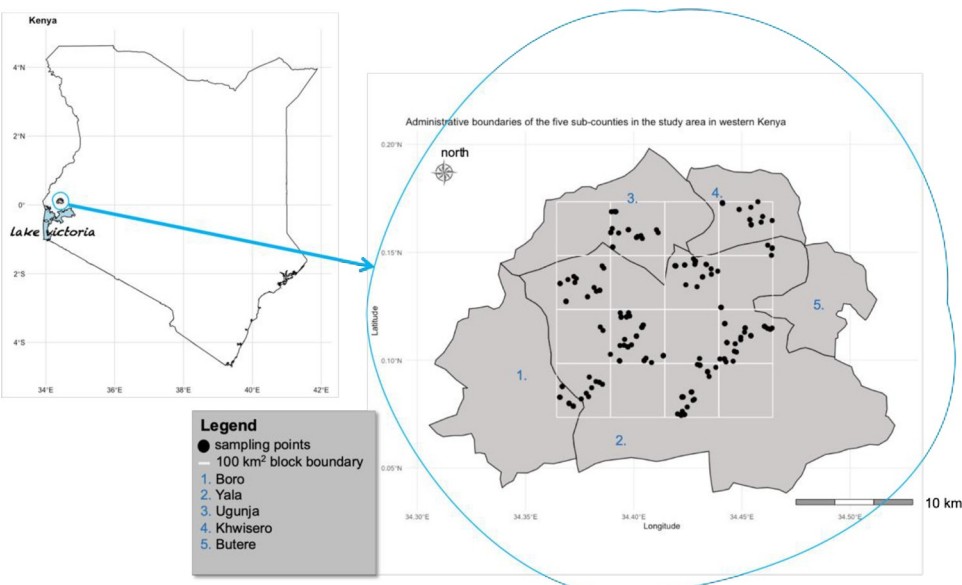

**Fig 1. Geographical location of study site within sub-county administrative boundaries of Kenya, where the population-based farm survey was conducted, and also shows location sampled plots within maize field (blue points).**

landforms vary, ranging from undulating hills and broad valleys, moderate lowlands, and swamps. Altitudes vary in the gentle landscape, between 1400 and 1500m above sea level. Bimodal rainfall patterns and a mean annual temperature of 20˚C, characterize this region [43]. The long rains occur from March to May, while short rain from October to December [43]. The major soil classes in the study area include Ferrasols (well-drained, moderately to very deep, clay soils) on the hills, Cambisols (well-drained, moderately deep, loamy clay soils) also the hills and Gleysols (poorly drained, shallow, sandy loam soils) on the plains [44].

Details on the study area, characteristic, and selection of the sites are provided by Ichami *et al.* [47]. The smallholder agroecosystem landscapes are characterized by subsistence farming systems, where farmers grow food crops such as maize (*Zea mays L.*), bananas (*Musa paradisiaca L.*), sweet potatoes (*Ipomoea batata L.*), and groundnuts (*Arachis hypogaea L*) [48]. The region is characterized by a high population density (500 persons per square kilometer) implying higher pressure on existing land for crop productivity within this smallholder agroecosystem.

## 2.2 Overview of the population-based farm survey approach

The PFS approach involved conducting a farm survey on several maize fields. The term "*population-based*" signified a population of a smallholder maize fields, within smallholder agroecosystem. A target population was defined as a representative sample population, drawn from a population of maize fields (with defined characteristics) that was evaluated. The target population forms the basis for making inferences about the nutrient diagnosis for the whole population of maize fields. The sequence of steps employed in this novel nutrient diagnostic approach is summarized in Fig 2, which provides guidelines for diagnostics, ranking, and mapping spatial patterns of nutrient limitations.

**2.2.1 Farm survey.** A farm survey was conducted using a Land Degradation Sampling Framework (LDSF) scheme [47], to obtain population estimates on soil and plant NPK

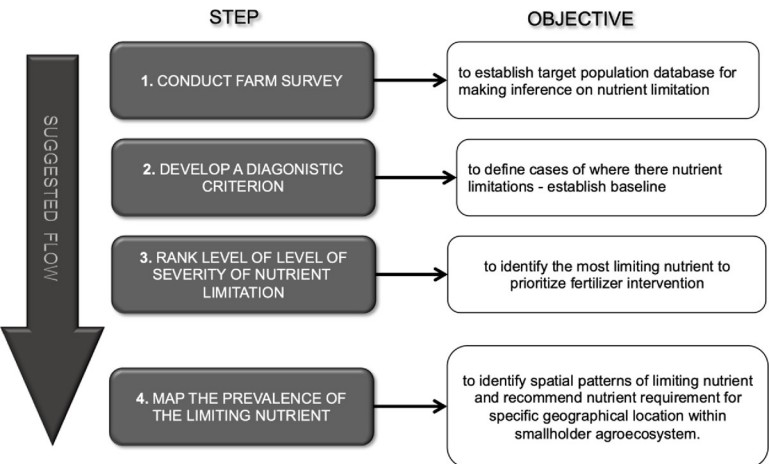

**Fig 2. Sequence of step used for implementing a population-based farm survey for nutrient diagnostics of maize fields within smallholder agroecosystem.**

nutrients status on targeted maize fields (Step 1, Fig 2). The farm survey was stratified into unfertilized maize fields, to capture variation in soil and plant information at different spatial scales. Details of the farm survey are described by Ichami *et al.* [47]. The LDSF is a stratified hierarchical design, which captured variability at different scale levels: block, tiles, sub-tiles, fields, and plots (Fig 3). A 100 km² square block was overlaid in the study area and sub-divided into 16 tiles (6.25 km²), and each tile was further sub-divided into 10 sub-tiles (0.25 km²). A total of 8 tiles and 32 sub-tiles were randomly selected, within which geographical coordinates were selected to represent maize fields. Three pairs of geographical coordinates were drawn for sampling in each sub-tile, for consecutive long and short rain seasons of 2013. Prior to conducting the survey, approval and consent were sought from the local government authorities and the smallholder farmers, owners of the farm where the samples were collected.

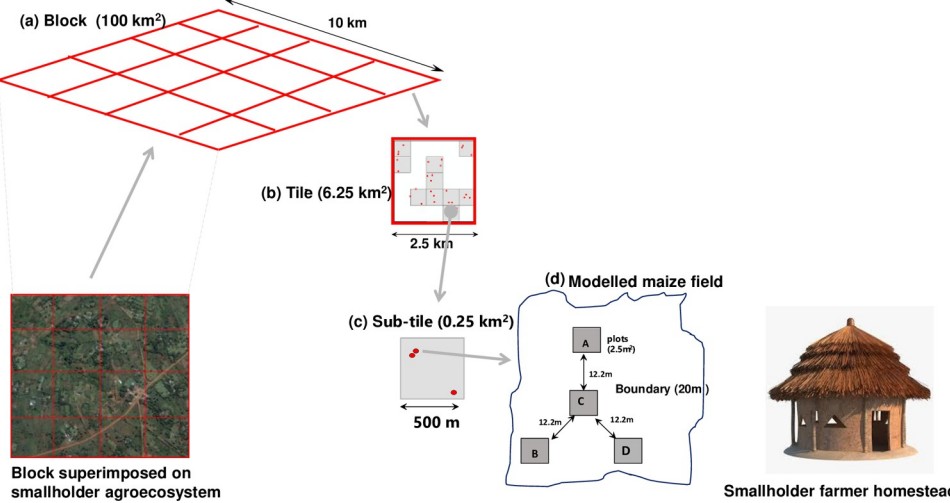

**Fig 3.** The Land Degradation Sampling Framework (LDSF) used for the sampling strategy (a) Square block measuring 100 km², (b) Tile 6.25 km² (c) Sub-tile with randomly selected maize fields (d) maize field (0.25 km²) with e) plots (A, B, C, D) in a Y frame layout, (12.2 m radius distributed uniformly at 360˚). The plots measure 2.5 m².

A maize field for sampling was identified using a global positioning system (GPS) device, by navigating through the smallholder landscapes. Unfertilized monocrop maize field, at the ear-leaf growth stage, that was well managed (e.g., free of weeds, pest and diseases), was selected for sampling. Only 2 nearby maize fields were sampled. The third field was not sampled, except for cases where the 2 prior selected fields, did not fulfill the aforementioned criterion. To capture a wide variation in nutrient concentration, a purposive sampling strategy was included, where a maize field was considered to be "*poor*" or "*good*", are purposely included for selection [49]. But only when they fell within the proximity of selected coordinates. Visual observation of the physiological maize attributes such as the stem height and basal diameter were used to discern the *poor* or *good* fields following Lafitte [50]. The defined good fields were those with healthy maize crops, while poor fields were associated with malnourished maize crops (Fig 4). This ensured the target population was sufficiently characterized.

Upon selection of a field, a Y frame was laid and a central plot was located first, by measuring 20 m from the main boundary, towards the center of the maize field. The main boundary was defined as the boundary located from the direction of smallholder farmers' homestead. Three plots were thereafter located, 12.2 m from the central plot, and distributed uniformly around it. Geographical coordinates, soil and plant nutrient concentration, and plant biovolume were determined for each plot (2.5 m$^2$).

Within the selected plot, soil sampling was achieved using a zig-zag pattern, where six soil samples were taken at 0 to 25 cm depth, using an Edelman soil auger (600 cm$^3$), to obtain a composite representative sample. Plant sampling was conducted by extracting three maize ear leaves samples, which were also selected using a zig-zag pattern, for nutrient analysis at 60 to 75 days after plant emergence (silking stage). The assumption was, at this growth stage, the maize tissue (ear-leaf) had an optimum concentration of nutrients.

Plant biovolume (BV) and grain yield (GY) were determined and represented crop responses in the study area. Plant biovolume was estimated using the basal diameter (BD) and

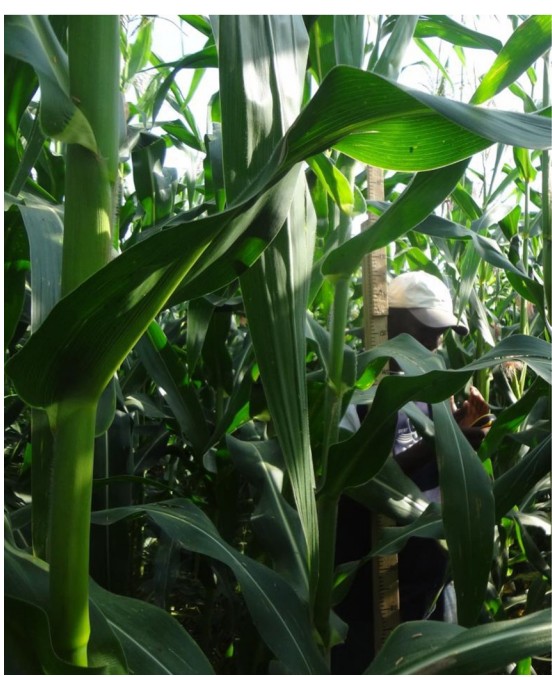 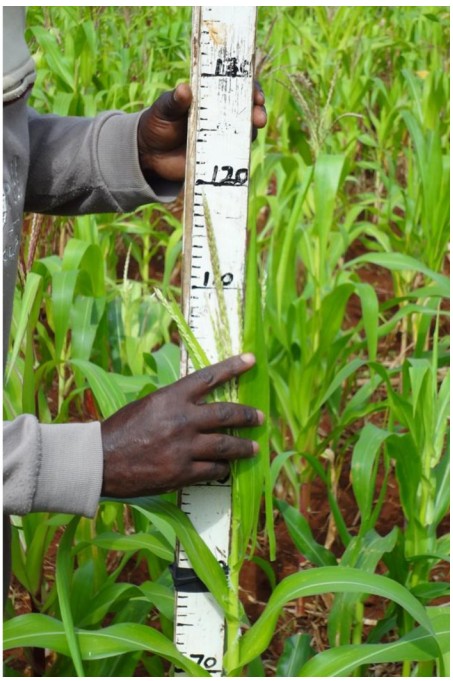

**Fig 4.** Photographs of a) good maize field, that is well managed and b) a poorly manage maize field, within the smallholder agroecosystems (Picture courtesy of *Stephen M. Ichami*).

height (H) of the maize plant, following Chomba *et al.* [49]: Eq 1.

$$BV(cm^3) = H(cm) \times \left(\frac{BD(cm)}{2}\right)^2 \pi \qquad (1)$$

The BD was measured in duplicate, two centimeters above the soil surface. Mean biovolume was estimated from measurements of all maize plants in the plot. Grain yield was measured from dry maize that was hand-harvested and the kernels removed and weighed (kg) at plant maturity (between 50–60 days after the silking stage).

**2.2.2 Spectral measurements of soil and plant samples.** All soil and plant samples were characterized for NPK nutrients by taking spectral measurements within the mid-infrared (MIR) region (4000–600 cm$^{-1}$). Preliminary preparation of soil samples involved air-drying and grounding, to pass through a 2 mm sieve to minimize variation due to moisture. While plant samples were washed under running tap water and rinsed with deionized water to remove contaminants, then oven-dried at 60° C, before grounding (< 1 mm) for spectral measurements.

Fourier-transform MIR spectrometer (FT-IR; Tensor 27, Bruker Optics, Karlsruhe, Germany) with a high throughput screening extension arm using a liquid Nitrogen cooled HgCdTe detector, was used to determine MIR diffuse reflectance. Prior to spectral measurements, soil samples were finely ground using a sample mill. Approximately, 0.05 grams of the fine sample were loaded and leveled into wells in aluminum micro-plates (A752-96, Bruker Optics, Karlsruhe) using a micro spatula, in four replicates (per sample). An empty well was used for reference readings, taken before each sample reading using an average of 32 scans. Absorbance was recorded at a spectral resolution of 4 cm$^{-1}$ zero-filled to 2 cm$^{-1}$ and a single spectrum was obtained for each sample. First derivative spectra with a smoothing gap of 3 points were used in all the spectral analyses [51].

Conventional laboratory methods were used to obtain reference data with 25% of the observations, which was used to develop calibration models. Total soil N was analyzed by dry combustion using a C/N analyzer [52]. Extractable P and K were determined using the Mehlich-3 extract [53], and an inductive coupled plasma optical emission spectrometer [54].

Plant samples were analyzed for total NPK tissue nutrient concentration following methods outlined by Okalebo *et al.* [55]. Total N was determined by sulphuric digestion followed by micro-Kjedhal distillation, while P and K were calorimetrically determined using vanadium molybdate after digestion with sulphuric acid [55,56], and P measured via flame photometry method [56].

**2.2.3 Spectral models and evaluation statistics.** Soil and plant NPK nutrient concentrations were predicted from spectral measurements data for all the samples using calibration models [37,57]. Wet-chemical reference values were calibrated to the smoothed first derivative spectra by conducting partial least square regression analysis using the "*soil. spec*" R package [51]. Soil and plant attributes were natural log- transformed for P and K to near normality distributions. The first-order derivative of the absorbance spectra over the MIR range was calculated using the Savitsky-Golay algorithm using the *soil.spec* R package, and was optimal for NPK [51]. The reliability and robustness of the calibration models were evaluated by the hold-out cross-validation procedure, using the coefficient of determination (R$^2$) and Root Mean Square Error of cross-validation (RMSECV), which were calculated using Eq 2.

$$R^2 = \frac{SSR}{TSS} \qquad (2A)$$

$$\text{RMSECV} = \sqrt{\sum_{i=1}^{N_p} \frac{(\widehat{y}cvi - y_i)}{N_p}} \tag{2B}$$

where, SSR is the sum square of regression, and TSS is the total sum of squares, $\widehat{y}_{cvi}$, and $y_i$ are the predicted and measured reference values respectively and $N_p$ is the number of samples tested. High $R^2$, and low RMSEV values were interpreted to have a good model fit.

The survey enabled the establishment of a database (Step 1, Fig 2), containing information on geographical coordinates, spectral measurement of soil and plant nutrient concentrations, GY and BV. Descriptive statistics for soil and plant nutrient concentrations were computed. The mean, maximum, minimum, standard deviation, coefficient of variation (CV), and confidence intervals (CI) values were determined. Density plots were used to test the normality of distribution of soil and plant variables. Skewed variables were transformed to achieve a near-normal distribution, before correlation analysis and geostatistical modelling.

**2.2.4 Development of nutrient diagnostic criterion.** The diagnostic criterion was based on soil test values for NPK nutrients (Step 2, Fig 2). Mean soil test values were compared to established critical nutrient values (Table 1) for NPK maize tissue concentration from the literature [58,59]. This formed the baseline soil test values for nutrient diagnostics. Maize plots were identified as deficient when soil test values were below the critical level of the nutrient value.

Maize ear-leaf nutrient NPK concentrations were subjected to regression analysis as a function of GY and BV. The relationships between maize tissue nutrient concentration and GY were evaluated following the Cate -Nelson method [63] and used to establish soil test cut-off values. The Cate -Nelson analysis enabled the calibration of soil test values to the study site. The analysis was also used to determine level where the addition of nutrients was likely to increase maize yield [63,64]. This was used to identify which maize fields were likely to respond to fertilizer application. These relationships were examined as a basis for establishing localized soil test values. The later values, with the corresponding maize yield, were used to define "*deficient*" and "*sufficient*" populations. Localized soil test values were identified by establishing a frequency distribution plot of two subpopulations, which were plotted as a function of NPK soil tests values. Critical soil test values were obtained based on the overlaps of the frequency distribution curves between the two subpopulations, at the upper and the lower 90% CI of two populations, respectively. This led to the next step of developing DRIS indices, for identifying important nutrients, and ranked their severity in terms of their deficiency.

**2.2.5 Ranking level of nutrient limitation severity.** The DRIS indices were computed as described by Beaufils [65]. The target population was divided into "*deficient*" and "*sufficient*" subpopulations, based on the maize yield values established through Cate and Nelson Analysis in Step 2 (Fig 2). The indices were used to determine relative degree of imbalance among nutrients in the study area [52]. The main objective of this analysis was to rank the level of nutrient limitations and imbalance (Step 3, Fig 2).

**Table 1. Established critical nutrient concentration values for maize crop.**

| Nutrient | Critical nutrient concentration (%) at deficiency level | Critical nutrient concentration (%) range sufficiency level | Source |
|---|---|---|---|
| Nitrogen | 3.00 | 4.00–6.40 | [59,60] |
| Phosphorus | 0.25 | 0.42–0.69 | [61] |
| Potassium | 2.00 | 3.50–5.00 | [59,62] |

Calculation of DRIS indices utilized norms derived from maize tissues nutrient compositions and corresponding yields, representing variability encountered by fields. Means and variances of maize tissue nutrient concentration were calculated for two subpopulations for each nutrient. The N/P or N/K or P/K were computed only for the *sufficient* subpopulations and then divided by the number of observations of each expression: Eq 3.

$$norm\ for\ nitrogen/phosphorus = \frac{N/P}{n} \tag{3A}$$

$$norm\ for\ nitrogen/potassium = \frac{N/K}{n} \tag{3B}$$

$$norm\ for\ phosphorus/potassium = \frac{P/K}{n} \tag{3C}$$

where, NPK refers to maize tissue nutrient concentration in percentage (%), and n is the number of observations in the *sufficient* subpopulations.

DRIS indices were calculated: using Eqs 4 and 5.

$$N\ Index = \left[\frac{f(N/P) + f(N/K)}{2}\right] \tag{4A}$$

$$P\ Index = \left[\frac{f(N/P) + f(P/K)}{2}\right] \tag{4B}$$

$$K\ Index = \left[\frac{f(P/K) + f(N/K)}{2}\right] \tag{4C}$$

$$where: f(N/P) = \left[\frac{N/P}{n/p} - 1\right]\frac{1000}{CV} \tag{5A}$$

when the actual value of N/P > n/p or

$$f(N/P) = \left[1 - \frac{N/P}{n/p}\right]\frac{1000}{CV} \tag{5B}$$

when the actual value of N/P < n/p

n/p is the mean (norm) value for N/P, and CV is the coefficient of variation for high-yielding populations. The other terms of *f*(N/K) and *f*(P/K) are derived in a similar way using the means, n/k for N/K and p/k for P/K, respectively in place of n/p. The interpretation of DRIS indices was based on the magnitude of their values, which sum to zero. The more negative a value appeared, the more a nutrient was ranked as more limiting and imbalanced.

**2.2.6 Mapping prevalence of limiting nutrients.** Spatial analysis was conducted to evaluate spatial variability of DRIS indices across the study area (Step 4, Fig 2). Spatial pattern of DRIS indices within the study area was analyzed by developing interpolation maps of DRIS indices using inverse distance weighted [66]. The spatial pattern described levels of nutrient geographic distribution within the study area. Maps for this study were developed using a "*ggplot2*" package [67]. This enabled identification of specific geographies where actual nutrient imbalance and limitation occurred.

Moran's (I) Index was computed and used to evaluate spatial auto-correlation by identifying the presence of clustered or dispersion patterns in NPK nutrient limitations, based on

DRIS indices [68]. The magnitude of Moran (I) index depicts levels of spatial clustering, with positive values taken to indicate clustering, and negative values showed spatial dispersion. To test the hypothesis that the spatial pattern occurrence of NPK nutrient is random within smallholder agroecosystems, the Local Moran's Index was computed using Eq 6

$$I = \frac{y_i - \overline{y}}{\delta} \sum_{j=1, j \neq i}^{n} [W_{ij}(y_j - \overline{y})] \tag{6}$$

where $\overline{y}$ is the mean value of y with the sample number of n; $y_i$ is the value of the variable at location $i$; $y_j$ is the value at other locations (where $j \neq i$); $\delta$ is the variance of z; and $W_{ij}$ is a distance weighting between $z_i$ and $z_j$, which can be defined as the inverse of the distance. Monte Carlo simulations for MI was conducted, to test the robustness of the prediction of the observed spatial patterns. The Moran I index were computed using the "*spdep*" R package [69].

## 3.0 Results

### 3.1. Farm survey database

The developed spatial database contained 256 plot observations from 64 maize fields that formed the target population for this study. Grain yield and BV represented crop response of unfertilized maize fields. At the time of grain harvesting, 7% of the sampled plots had been harvested by farmers. The final database contained spectral information on soil and maize ear-leaf tissue NPK nutrient concentrations, GY and BV with 256 records across 11 variables (latitude, longitude, altitude, soil N, soil P, soil K, tissue N, tissue P, tissue K, GY, biomass), including missing values of GY.

Spectral calibration models for NPK gave good fits with cross-validated $R^2$ values of 0.88, 0.68, and 0.74, respectively (Table 2). The robustness of the calibration models varied, as shown by the different fit of $R^2$ and RMSECV values. Soil test N concentration had the lowest RMSECV of 0.09, compared to Extr. K, which had a higher value of 0.44. However, the poor prediction was evident for plant tissue phosphorus (see S2 Fig). Nitrogen prediction was robust compared to Extr. P and K.

Predicted means for soil N varied from 0.06% to 0.36%, while Extr. P and K had a median of 17.2 mg kg$^{-1}$ and 4.6 cmol$_c$ kg$^{-1}$, respectively (Table 3). Low nutrients concentration, characterized the soils of study area, as exhibited by means of soil N, Ext. P, and K. Cases of soil NPK nutrient deficiencies were evaluated based on the critical soil values of 0.2%, 10 mg kg$^{-1}$, and 3 cmol$_c$ kg$^{-1}$, for NPK, respectively [55,62]. Nutrient deficiency cases were prevalent in 57%, 61% and 43% of observations, for soil N, Extr. P and K, respectively.

**Table 2. Mid-infrared calibration model statistics that predicted soil and plant nutrient concentrations of the study area.**

| Nutrient concentration | Coefficient of determination (R$^2$) | Root Mean Square Error of Cross Validation (RMSECV) |
|---|---|---|
| | **Soil samples** | |
| Total Nitrogen | 0.88 | 0.09 |
| Extractable Potassium | 0.74 | 0.44 |
| Extractable Phosphorus | 0.68 | 0.40 |
| | **Maize ear-leaf samples** | |
| Nitrogen | 0.84 | 0.08 |
| Phosphorus | 0.84 | 0.16 |
| Potassium | 0.80 | 0.12 |

**Table 3. Soil properties, maize ear leaf total tissue nutrient concentration and crop response variables of unfertilized maize plots across a smallholder agroecosystem in western Kenya.**

| Nutrient | n | Minimum | Maximum | Median | Mean | Standard deviation | Coefficient of variation | Confidence Interval mean (95%) |
|---|---|---|---|---|---|---|---|---|
| **Soil properties** | | | | | | | | |
| Total N (%) | 245 | 0.06 | 0.36 | 0.14 | 0.15 | 0.04 | 25% | 0.01 |
| Extr. Potassium (mg kg$^{-1}$) | 256 | 8.19 | 107.21 | 17.22 | 21.02 | 12.82 | 61% | 1.58 |
| Extr. Phosphorus (cmol$_c$ kg$^{-1}$) | 256 | 0.12 | 6.48 | 4.61 | 4.24 | 1.49 | 35% | 0.18 |
| **Maize ear-leaf samples** | | | | | | | | |
| Nitrogen (%) | 220 | 0.12 | 0.40 | 0.22 | 0.23 | 0.06 | 25% | 0.01 |
| Phosphorus (%) | 220 | 0.57 | 2.76 | 1.84 | 1.80 | 0.45 | 25% | 0.06 |
| Potassium (%) | 220 | 0.00 | 3.81 | 1.86 | 1.68 | 0.70 | 42% | 0.11 |
| **Crop responses** | | | | | | | | |
| Grain yield (Mg ha$^{-1}$) | 203 | 0.08 | 11.28 | 3.20 | 3.54 | 1.90 | 54% | 0.23 |
| Plant bio-volume (cm$^3$) | 203 | 31.00 | 392.91 | 161.00 | 170.38 | 73.33 | 43% | 9.10 |

n = number of observations, Extr. = Extractable.

The maize fields were characterized by variation in soil and plant tissue nutrient concentration, as well as crop responses. High variation occurred in Extr. P with a CV value of 61% compared to a corresponding value of 25% for P concentration for the maize ear-leaf tissue (Table 3). Soil N exhibited low variation (CV = 25%). A similar trend was observed for N and P ear-leaf tissue nutrient concentration, with CV values of 25% for both. Grain yield and BV displayed high variability with CV values of 55% and 40%, respectively. The high CV statistics indicate that the farm survey captured variability in crop response and nutritional status of the study area. Outlier (11) that displayed very high N concentration were expunged in the preceding analysis.

Density plots for soil N, GY, and BV displayed a near normal distribution. Extr. P and K were negatively skewed, indicating the presence of low values amongst the target population. Skewed variables were transformed to natural log (*ln*) values, to attain approximate normal distribution, as required in the subsequent steps of statistical evaluation of the population-based farm survey approach.

## 3.2. Diagnosis of limiting nutrients

Correlation analysis indicates a linear increase of 67% and 30% of GY and BV, respectively, with increasing maize tissue P concentration (Fig 5). A generally observed trend was an increase in maize tissue nutrient concentration, which corresponded to 5.37 Mg ha$^{-1}$, and 240 cm$^3$ increase for GY and BV for a unit increase in tissue P, respectively (Fig 5B, 5C and 5E). Tissue N had a positive significant relationship ($p = 0.018$, $R = 0.22$) with BV, as would be expected at the silking stage (Fig 5B). The relationship between GY and N contrasted that of BV and N, which was negative and poorly correlated ($p = 0.54$, $R = -0.085$). Similar non-significant negative relationship was observed between K and BV ($p = 0.26$, $R = -0.11$). Even though there were weak correlations as shown by low $R$ values ($< 0.5$), tissue N concentration had the strongest significant relation ($p = 0.018$, $R = 0.22$) with BV compared to the relationships between GY with P and K tissue nutrients (Fig 5). Poor relations can be attributed to low nutrient concentration in the soil of the study area, nutrient imbalance, and deficiency in micronutrients that affect nutrient uptake by the plant [36,70,71]. The results indicate the poor capacity of plant tissue to predict maize grain yield.

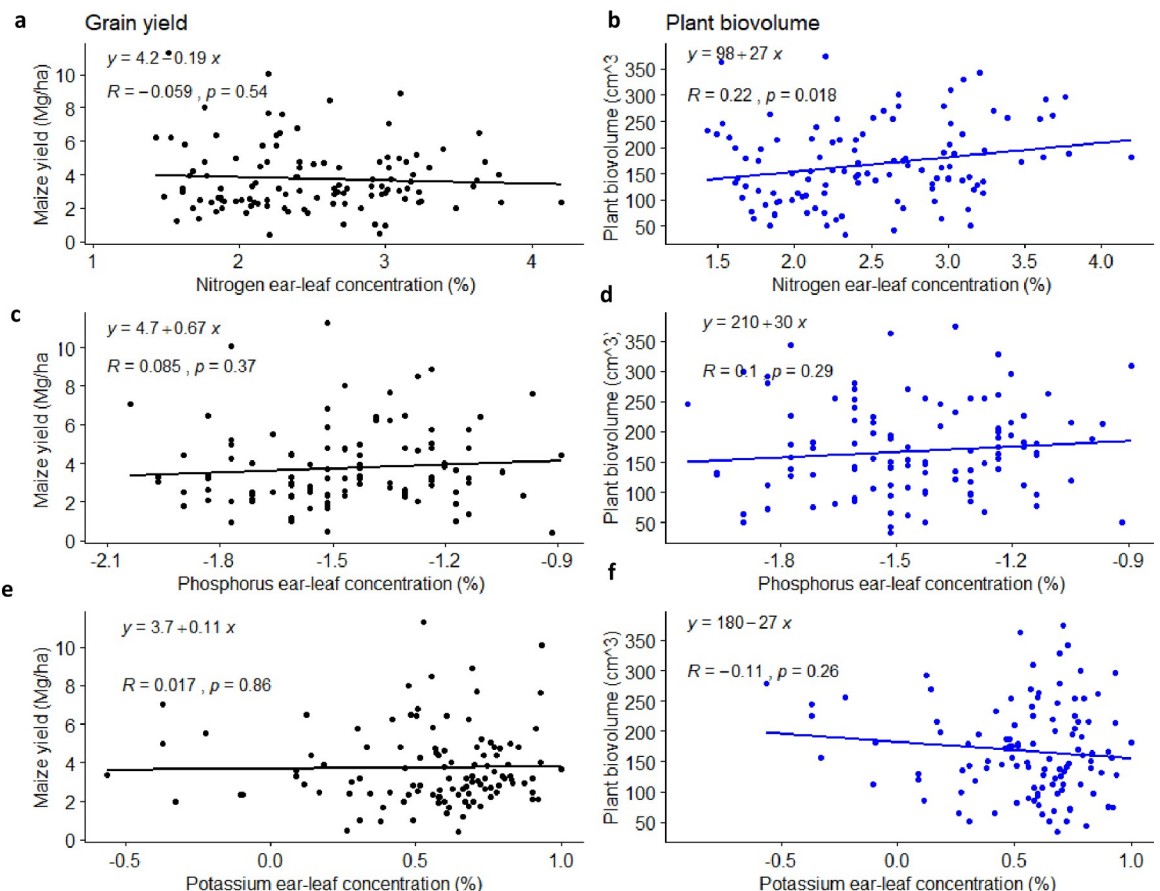

**Fig 5.** Relationship between grain yield (Mg ha$^{-1}$) and plant biomass (cm$^3$) as function of maize tissue total nutrient concentration for; a, b) nitrogen (%), c, b) phosphorus (%), and e, f) potassium (%).

The significant relationship between BV and maize ear-leaf tissue concentration for N was indicative that they could be inferred and used to measure nutrient limitations. Therefore, established critical nutrient value for total N formed a basis for dividing the target population into a "*deficient*" and "*sufficient*" subpopulation. The deficient subpopulation constituted 46%, a representation of poor maize fields in the study area. The two subpopulations formed the basis for establishing localized soil cut-off values for the study area.

Established soil cut-off values were used to determine the prevalence of soil nutrient limitation for the study area (Table 1). The target population was characterized by soil N deficiency in 67% of the sampled field, and 54% in phosphorus, while 37% was observed for potassium. Approximately 15% of the maize fields had N levels in the range of 0.1–0.25%. Therefore, it was clear that soil N was a major factor limiting maize growth in the study area.

Fig 6 shows results of Cate–Nelson analysis used to determine and calibrate (localize) the critical soil test value for NPK. The soil critical values were 0.13% for N, 18.3 for Extr. P and 5.3 cmol$_c$ kg$^{-1}$ for Extr. K, below which crop response was likely. These critical soil test values correspond to the GY yield of 3.3 Mg ha$^{-1}$, 2.0 Mg ha$^{-1}$, and 5.9 Mg ha$^{-1}$ for NPK, respectively. Based on this analysis, the target population was divided into responsive and non-responsive categories. Quadrants II and IV represented the responsive plots while I and III represented the non-responsive. Result indicates 76%, 61% and 79% of the maize plots would be responsive to NPK applications, respectively. The Cate-Nelson analysis for partition the target population

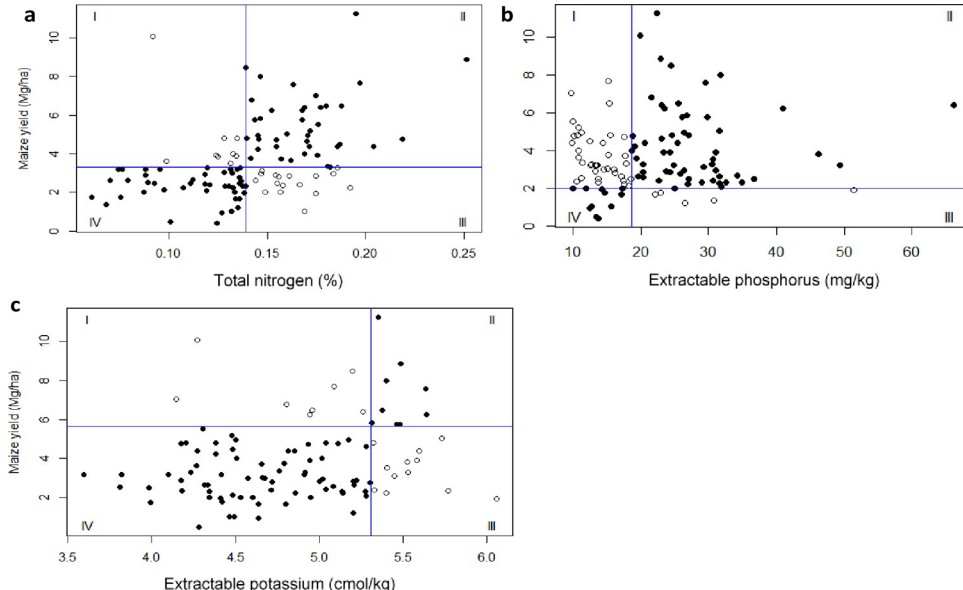

**Fig 6.** Soil test critical values with the Cate-Nelson method analysis for **a**) Nitrogen, **b**) Phosphorus and **c**) potassium. The intersection of blue lines, the vertical and horizontal line of the x, and y represents critical values for soil test values beyond which crop response is unlikely. The black dots are characterized as responsive plots while the white dots are non-responsive plots.

was robust as indicated by the ratio between responsive and non-responsive maize plots with $r^2$ values of 0.76 for N, 0.61 for P and 0.68 for K.

Fig 7 presents frequency distribution plots for soil tests values for total N, Extr. P and K. Localized soil cut-off values were 0.097% for soil N, 12.5 mg kg$^{-1}$ for Extr. P and 4.5 cmol$_c$ kg$^{-1}$ for Extr. K. There was no clear separation of *deficient* and *sufficient* subpopulations for total N and Ext. K (Fig 7B and 7F). In the frequency distribution plot for Extr. P, it performed better for the separation of the *deficient* and *sufficient* populations (Fig 7D). Result also suggests soil cut-off values can be obtained using farm survey data, since there were no significant differences in magnitude between developed farm survey soil cut-off values, compared to published critical soil test values [55].

### 3.3 Ranking limiting nutrients

The target population consisted of 256 observations, and were divided into *deficient* and *sufficient* populations, which comprised of 115 and 141 observations, respectively. These subpopulations formed basis for computing DRIS indices. Calculated variance ratio of N/P, N/K and P/K were significantly different ($p < 0.01$). The significance of variance provided evidence on the validity of the assumption used in separating the two aforementioned subpopulations. *Deficient* subpopulation showed high values of standard deviation (0.18) and CV (34%) for N compared to *sufficient* subpopulation.

The DRIS indices varied widely, from -35.86 to 36.87 for N for the study area. Mean DRIS indices were -6.3 for N, -13.5 for P, and -2.1 for K (Fig 8). Extractable P was ranked as the most limiting nutrient compared to N and Extr. K. The relative ranking of the limiting nutrient, in ascending order from the most limiting was phosphorus > nitrogen > potassium. The results imply that fertilization of the maize crop in the study area may prioritize fertilization of phosphorus since it is the most limiting nutrient.

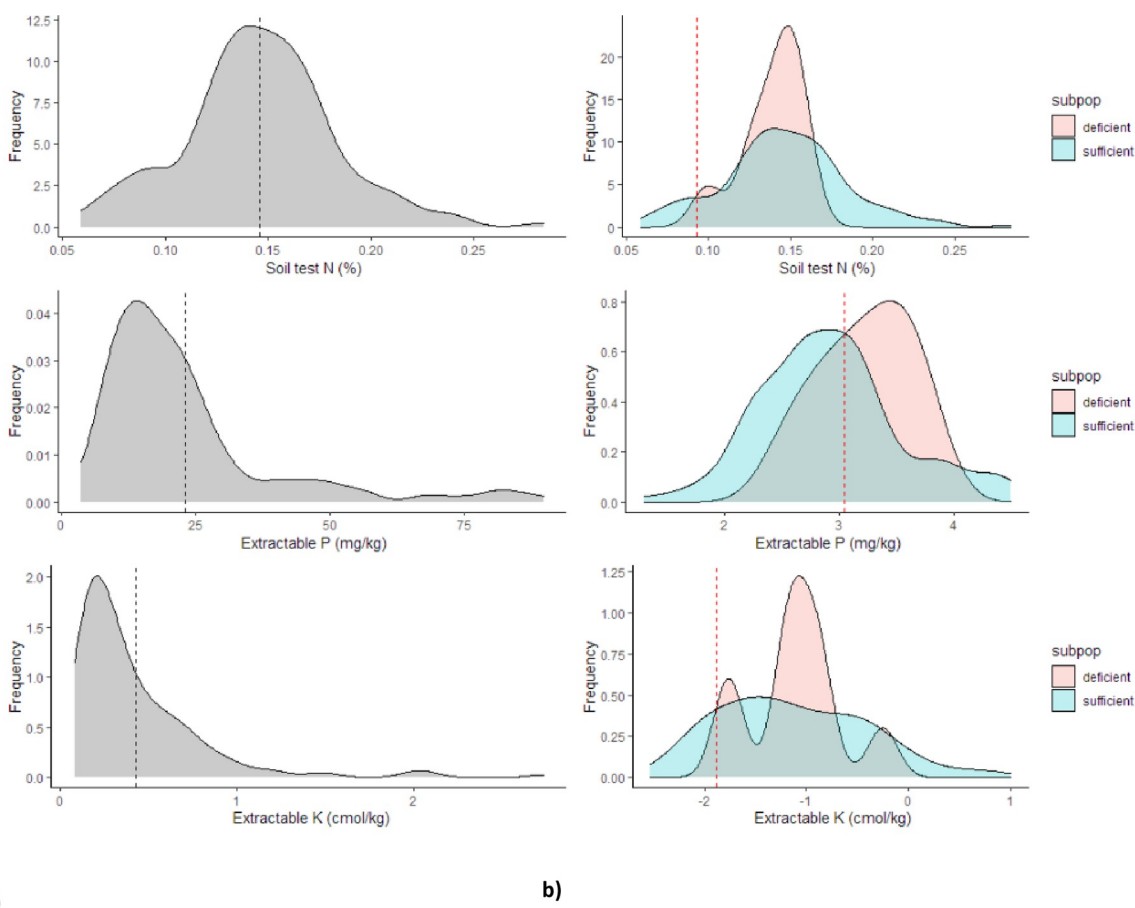

a)                                                                    b)

**Fig 7.** Density plots for soil test values **a**) for nitrogen, **c**) for phosphorus and **e**) potassium with corresponding to frequency distribution plot of "*deficient*" and "*sufficient*" sub-population for establishing localized soil cut-off values for **b**) nitrogen, **d**) for phosphorus, and **f**) potassium. The dashed red lines represent the soil cut off values, while the black one represents the mean soil test values of target population.

### 3.4 Map prevalence of limiting nutrients

Our study identified spatial patterns for NPK nutrient limitation using maps for DRIS indices for the study area (Fig 9). The bubble maps were used to visualize nutrient limitation across specific maize fields occurring in the region. The color gradient intensity represents the level of nutrient limitation which corresponded to proxy values, from severe ($<$ 0.5 blue hues) to optimum ($>$ 5 yellow hues). There was a variation in nutrient limitation patterns across the study area. For example, severe K nutrient limitation were observed in Yala sub-county. However, N was diagnosed as limiting in 64% of the maize fields, with severe, low and moderate nutrient limitations within the same sub-county. Gradient in P limitation was evident as indicated by the blue hue in Yala sub-county, with only seven maize fields diagnosed as optimum for P nutrient status. We observed nutrient limitation everywhere in the smallholder agroecosystems, and many maize fields ($>$ 50%) had moderate and severe nutrient limitations (see S4 Fig). The display of NPK geographical patterns of nutrient limitations meant that further evaluation of spatial autocorrelation using Moran Index (MI) was necessary to ascertain whether clustering of nutrients was present in the study area.

Positive Moran Index (MI) values were observed with values $>$ 0.20 (Table 4). A hypothesis that the spatial occurrence of NPK nutrient limitation is random, within smallholder

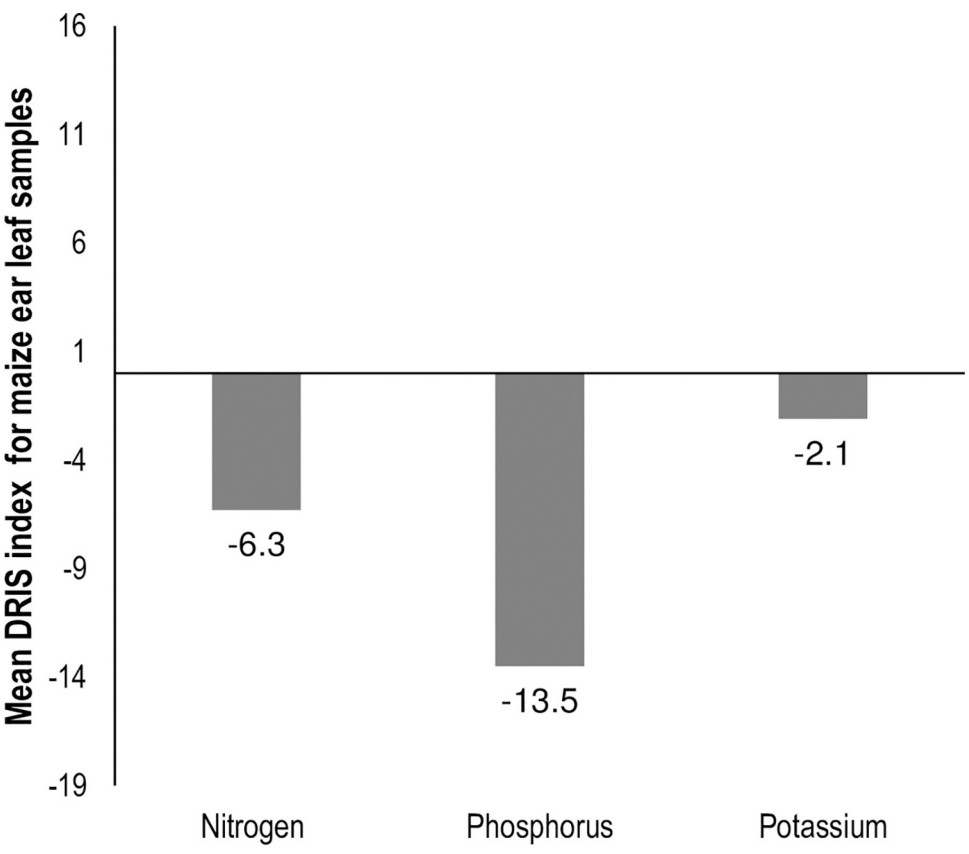

**Fig 8. Mean DRIS indices for nitrogen, phosphorus and potassium for maize fields across the study area.**

agroecosystem was rejected since MI for NK was significantly positive ($p < 0.001$). Thus, the MI indicated clustering in NPK nutrient limitation for the study area. Specifically, we find NPK nutrients displaying a clustering pattern, where multiple deficiencies of NPK occurring in the same geographical location, within the study area. Monte Carlo simulation for the MI using 599 interactions displayed significant values for N and K, indicating evidence of the robustness of the results.

## 4.0 Discussion

The main purpose of this study was to develop a Population-Based Farm Survey approach for nutrient diagnostics in smallholder agroecosystems. The novelty of the PFS approach comes from the combination of technologies that generate synergy between them. The critical nutrient concentration concept was established by Cate and Nelson [63], the DRIS approach [72], and spatial mapping tools [73,74]. The combination of these technologies generates significant synergisms and has demonstrated potential for the diagnosis of nutrient limitations in heterogeneous smallholder agroecosystems. These tools were incorporated in this current approach, following the principle of LHS of rigorous sampling and case definition, to estimate nutrient limitation at specific geographical niches within agroecosystem [7].

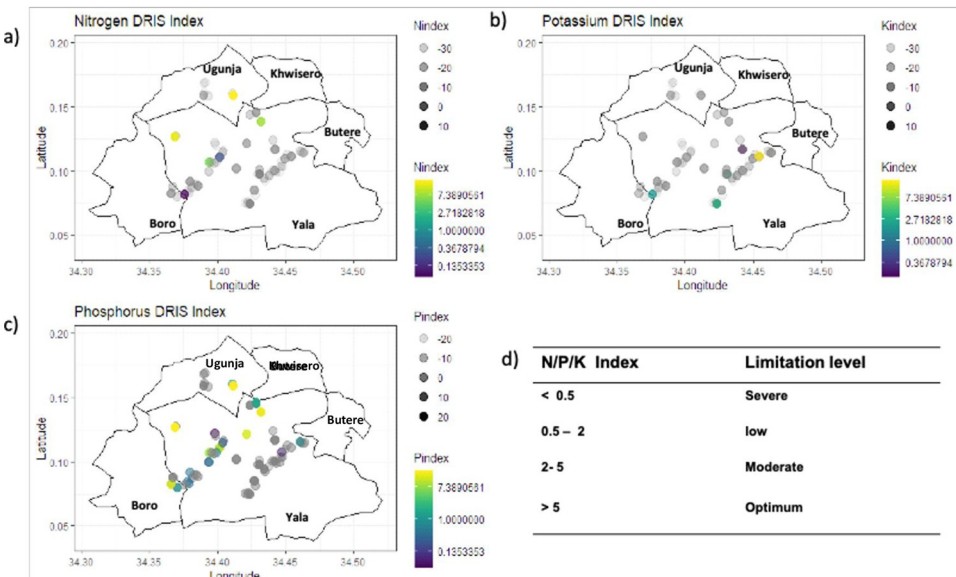

**Fig 9.** Maps showing the geographical location of maize fields and the spatial pattern of the intensity of nutrient limitations across the study area based on DRIS indices values for **a)** nitrogen, **b)** potassium and **c)** phosphorus.

Farm survey employed MIR spectral analysis to predict nutrient limitations for NPK for the study area (Table 2). The spectral calibration prediction model was robust but varied between soil and the maize ear-leaf tissue nutrient concentration for NPK. The fundamental vibrations of molecules in soil and plant materials were found in the MIR spectral region, with distinct spectral signatures displayed. This is attributed to the strong absorption of overtones by hydroxyl ions [37]. The best prediction was observed for soil N, with $R^2$ of 0.88 (Table 2), and within ranges reported by Yang *et al.* [75], followed by Extr. P with $R^2 = 0.74$, similar to those reported by Maleki *et al.* [76]. This calibration were better compared to those reported by Janik *et al.* [77] for Extr. P. The $R^2$ ranged from 0.60 to 0.90, and their (RMSECV) were satisfactory for diagnosis of nutrient deficiencies [78]. Even though previous studies showed poor prediction for phosphorus [77,79], it is clear that MIR spectra contain useful information about NPK nutrient limitations in general. These findings demonstrate the potential of using MIR spectra for the assessment of nutrient deficiencies and can be embedded in the implementation of PFS approach for smallholder nutrient management strategies [57,80]. The main advantage of this technique is the reduction of cost and rapidness, especially when a target population has many soil and plant samples, to be analysed for large areas [81].

Crop response, soil nutrient concentration, and maize tissue samples displayed a high degree of variability as evidenced by the CV values (Table 3). The observed variability was attributed to the difference in the inherent soil fertility, due to the influence of topography [82], management aspects [83], and soil types [84]. These results provide evidence of high variability in NPK nutrients across the smallholder landscape. This can be explained by the effects of climate, soil and historical management in the study area [85,86]. The finding is in

**Table 4. Moran index for the for maize fields in a smallholder landscape in Western Kenya.**

|  | **Nitrogen** | **Phosphorus** | **Potassium** |
|---|---|---|---|
| Moran Index | 0.40 | 0. 23 | 0.42 |
| *p* value | 0.0003 | 0.31 | 0.01 |

agreement to those reported by Tittonell *et al.* [12] in smallholder farms of western Kenya. Thus, a spatially-explicit approach for diagnosis of limiting nutrients, as an amelioration strategy for nutrient management for the smallholder agroecosystems is necessary. As a result, it would lead to a site-specific nutrient management strategy for the study area (Fig 2).

Soil and plant relations, although weak, were used to establish soil cut-off values useful in defining cases where nutrients were limiting (Figs 5–7). Developed soil cut-off values fell within the established ranges [55,87]. For example, Adeoye and Agboola [87] reported values ranging from 10 to 16 mg kg$^{-1}$ for phosphorus, and 0.6 to 0.8 cmol$_c$ kg$^{-1}$ for K in Nigeria. Based on developed soil cut-off values (Fig 7), diagnoses of NPK limitation were established in 67%, 54%, and 37% of sampled maize fields studied. There was no significant variation in the field (percentage) diagnosed as deficient using the already established (baseline) soil test values derived from the literature (Table 2). Additionally, our results indicate NPK were limiting maize production in the study area, and agree with those reported by Kihara *et al.* [88] in western Kenya, from nutrient omission trials. Although similar conclusions are reached with our findings, a striking difference with the PFS approach was the sampling strategy. We used farm survey data sampled from 256 plots, and in both studies, NPK were found to be limiting. However, the farm survey approach used 256 plots, spread across the study area, compared to 32 plots for nutrient omission trials within the same region [88]. The advantage and robust of PFS approach, is the calibration to actual farm conditions and development of local soil test value using the Cate and Nelson analysis to discern whether the maize field has excess or low NPK nutrients concentration (Fig 7). This approach was useful in predicting of non-responsive maize fields to fertilizer application, a problem that needs attention in smallholder agroecosystems. Thus, fertilizer investment strategies can be focused on the *deficient* maize plots to "flatten the curve".

The frequency plot of the "*deficient*" and "*sufficient*" subpopulation as a function of soil test values displayed a near-normal distribution for soil N and Ext. P (Fig 6B and 6D). The observed distribution can be explained by high soil heterogeneity [89], soil disturbances through ploughing [90], and low concentration of phosphorus, fixation of P, and potassium values for the study area [5,34]. Even though soil test values provided a criterion for defining cases of NPK limitations within the study area, they present disadvantages for nutrient diagnostic, when used in isolation [91,92]. Hence the need of incorporating maize tissue nutrient concentration. The relationships between measured ear-leaf tissue nutrient concentration and crop response were very tenuous (Fig 5), similar to the observations made in previous studies [32,35]. This can be attributed to the maize growth and its nutrient uptake rate in the field, which depend on many environmental factors such as moisture, soil nutrients concentration, and their interactions, that varied (Table 3) across the maize fields [18]. The growing plant integrates all these soil factors and is the best measure of true nutrient availability of the unfertilized maize fields [93].

The synergy of the current approach was first evident by evaluating nutrient concentration in both maize tissues and soil samples (Figs 5–7) where established critical nutrient concentration test values formed the basis for defining cases limiting nutrient. A positive correlation between maize tissue N concentration and plant biomass was observed (Fig 5B), which could be explained by high nitrogen uptake at the silking stage as expected [94]. Observations of positive and significant correlations between N concentration in maize leaves and GY are in line with those of Bak *et al.* [95]. Poor correlation between GY and P concentration (Fig 5) observed in the study could be attributed to low levels of P in the soil [25]. Poor relation between Extr. K and grain yield could be attributed to the poor prediction of the magnitude of GY (Fig 5C). The results differ from those reported by Clover *et al.* [96] who found good relations between potassium and grain yield, but in fertilized maize fields. The utility of critical nutrient concentration in maize tissue provides a synergy since these values guided

establishing *deficient* and *sufficient* sub-populations, on whose basis the DRIS indices for diagnosis of limiting nutrients were developed. This study, therefore, supports our argument that soil test values or maize tissue level are better used together rather than in isolation. These are complementary tools, which can used as a criterion of defining cases where nutrient limitation occurs within smallholder agroecosystems. The approach also demonstrates the development of localized soil cut off values.

The DRIS indices showed varying values for NPK nutrient limitation in the study but were within ranges reported by Nziguheba *et al.* [43] for maize crops (Fig 8). Potassium was ranked as the least limiting nutrient compared to phosphorus was the most limiting nutrient. Thus, fertilizer recommendations of N-based fertilizers with high portions of P and low K would be appropriate. The DRIS indices were used because they were not affected by differences in growth stages of maize crops [72]. The indices demonstrate the incorporation of plant nutrient concentration in nutrient diagnostics and bring the second synergy into this current approach. The DRIS technique was found to be advantageous since it took into account the nutrient status of the whole maize plant [97]. The results of DRIS were associated with uncertainty since the indices cannot be used to make fertilizer recommendations, although they provided substantial additional information on nutrient limitation ranking [58,98]. Thus, the indices should be accompanied by soil samples taken from the same area, hence the need for synergies for these nutrient diagnostic methods.

Diagnosis of limiting nutrients and ranking was established in the aforementioned section. However, the results do not explicitly explain, the spatial distribution of nutrient limitation of the study area. Thus, a geostatistical technique was employed to evaluate geographical distribution, which brings in the last synergy of the population-based farm survey approach.

A hypothesis that the spatial occurrence of NPK nutrient limitation is a random pattern within smallholder agroecosystem was tested and rejected since the MI for NPK nutrients were positive (Table 4). Test of significance for MI values for N displayed a significant clustered distribution ($p < 0.001$, MI = 0.40) (Table 4). The spatial pattern of P and K did not appear significantly different from a random distribution for this region (Table 4). The result for N clustering conforms to those by Panday *et al.* [99] who found significant clustering for N in smallholder farms of Nepal. Clustering may be taken as an indication of the occurrence of NPK limitations in one location, which requires a holistic approach for nutrient management for different geographies niches. The analysis of the spatial pattern of DRIS indices provided synergy through evaluation of geographical location of nutrient limitations. In this way, nutrient management strategies could be implemented using the spatial distribution maps as a guide for identifying occurrence of nutrient limitation in specific geographical niches within agroecosystems. The clustering pattern can be explained by differences in soil characteristic patterns, which are complex due to the topography of the area [100,101].

The spatial maps (Fig 9) depicted the status of NPK contents in different geographical niches across the study site most of which displayed their deficiencies, as indicated by negative DRIS indices (Fig 9). The deficiency could be explained by different historical management practices (no fertilizers were applied) that have influenced inherent soil properties [4,5,102]. This finding implied that by making reference to the DRIS indices simulated maps, NPK fertility status could be assessed to guide site-specific fertilizer application. Normally, low nutrient values require a relatively higher amount of fertilizer application; therefore, these maps may lead to a better understanding of existing nutrient limitation, allowing sustainable maize productivity. This research thus sets a precedent for upscaling the population-based farm survey approach of nutrient limitations in other smallholder farming systems, with similar conditions. Its implementation would enhance strategies for site-specific fertilizer recommendations for smallholder agroecosystems.

## 5.0 Conclusion

The study developed a population-based farm survey approach for the diagnosis of limiting nutrients for smallholder agroecosystems in western Kenya. Soil test values for N (0.01%), Extr. P 12.2 mg kg$^{-1}$ and K (4.5 cmol$_c$ kg$^{-1}$) were developed from quantitative soil and plant relationships and then used to define cases of nutrient deficiencies. Deficiency of nitrogen, phosphorus and potassium limit maize production in the study area. Spatial maps for nutrient limitations were developed, which identified the occurrence of nutrient limitations in specific geographies for the study area. This study demonstrated that site-specific diagnosis of nutrients can be implemented in this region and other regions with similar characteristics. This may lead to effectiveness and optimize fertilizer use recommendations in the region. Therefore population-based farm survey approach is an effective diagnostic approach for exploring spatial variability of soil nutrients and can be upscaled for future use in similar smallholder agroecosystems. Future research efforts need to evaluate the optimum number of observations in a target population with respect to the total area to be covered by the farm survey.

## Supporting information

**S1 Fig. Spectral measurement results for the calibration models and predicted results for nitrogen from the farm survey data of the study area.**
(DOCX)

**S2 Fig. Spectral measurement results for the calibration models and predicted results for phosphorus from the farm survey data for the study area.**
(DOCX)

**S3 Fig. Spectral measurement results for the calibration models and predicted results for potassium from the farm survey data for the study area.**
(DOCX)

**S4 Fig. Stimulated maps for NPK DRIS indices displaying spatial pattern of nutrient limitations across the study area.** The green points represent the maize fields. The maps were interpolated using inverse distance averages.
(DOCX)

## Acknowledgments

The research was technically supported by Wageningen University through a sandwich Ph.D. program Tropical Soil Biology and Fertility Institute (CIAT-Kenya) and World Agroforestry Centre (ICRAF) fellowship programs in collaboration with the University of Nairobi. We acknowledge initial comments on the concept of this paper by Prof. Ellis Hoffland and Dr. Jetse Stoorvogel of Wageningen University. The unlimited cooperation and support extended by African Plant Nutrition Institute (APNI) through Dr. Shamie Zingore in carrying out the field research work in western Kenya are gratefully appreciated. We also acknowledge the support of the CGIAR research program on 'Water, Land and Ecosystems'.

## Author Contributions

**Conceptualization:** Stephen M. Ichami.

**Data curation:** Stephen M. Ichami, Andrew M. Sila.

**Formal analysis:** Stephen M. Ichami, Andrew M. Sila, Fredrick O. Ayuke.

**Funding acquisition:** Keith D. Shepherd.

**Investigation:** Stephen M. Ichami, Keith D. Shepherd.

**Methodology:** Stephen M. Ichami, Keith D. Shepherd.

**Resources:** Stephen M. Ichami.

**Software:** Stephen M. Ichami.

**Supervision:** George N. Karuku, Fredrick O. Ayuke, Keith D. Shepherd.

**Visualization:** Stephen M. Ichami, Andrew M. Sila.

**Writing – original draft:** Stephen M. Ichami, Andrew M. Sila.

**Writing – review & editing:** Stephen M. Ichami, George N. Karuku, Andrew M. Sila, Fredrick O. Ayuke, Keith D. Shepherd.

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
