## [Decision Letter · Decision Letter 0]

19 Jul 2021

PONE-D-21-15324

Spatially approach for diagnosis of yield-limiting nutrients in smallholder agroecosystem landscape using population-based farm survey data.

PLOS ONE

Dear Dr. Muhati,

Thank you for submitting your manuscript to PLOS ONE. After careful consideration, we feel that it has merit but does not fully meet PLOS ONE’s publication criteria as it currently stands. Therefore, we invite you to submit a revised version of the manuscript that addresses the points raised during the review process.

We look forward to receiving your revised manuscript.

Kind regards,

Elisabeth Bui

Academic Editor

PLOS ONE

Journal Requirements:

Additional Editor Comments (if provided):

The paper has improved and is close to being acceptable for publication. Please address the comments made by the two referees first; many editorial corrections need to be made, starting with the title "Spatially explicit?.." or "Spatial approach...".

Reviewers' comments:

Reviewer's Responses to Questions

**Comments to the Author**

1. Is the manuscript technically sound, and do the data support the conclusions?

Reviewer #1: Yes

Reviewer #2: Partly

2. Has the statistical analysis been performed appropriately and rigorously? 

Reviewer #1: Yes

Reviewer #2: Yes

3. Have the authors made all data underlying the findings in their manuscript fully available?

Reviewer #1: Yes

Reviewer #2: Yes

4. Is the manuscript presented in an intelligible fashion and written in standard English?

Reviewer #1: Yes

Reviewer #2: Yes

5. Review Comments to the Author

Reviewer #1: The manuscript has been tremendously improved and made better. However, I will suggest that authors move away from trying pitch soil health to soil nutrients. Dealing with nutrient deficiencies alone as described in Line 58 - 60 is not the ultimate solution for poor soil health. And so authors could include "with other improved agronomic practices and biological processes" which are also very important in making 'judicious fertilizer investment' work.

Authors could also bring clarity in "heterogeneity" as used in their context since they specifically deal with nutrients variation. Soil heterogeneity as used here is too broad - Line 61 - 66. 'site-specific' nutrient diagnostics target resolving issues with nutrient variation and not soil heterogeneity and therefore cannot be replaced with it.

Line 177: change 'captured' to capture

Line 290: remove the ',' after 'identify'

Figure 9: Authors should make the labelling of the maps consistent. A and B have all the labels except C. In addition, location of Butere has been displaced in C. Authors should consider dropping Khwisero and Butere from the map since no data are recorded in these locations.

Reviewer #2: The paper is publishable due to the novelty of the approach. In some instances more care needs to be taken to interpret the data presented in figures accurately within text. There are several grammatical errors that can be easily fixed with another proof read. I have noted some of these as examples, but not all.

6. PLOS authors have the option to publish the peer review history of their article (what does this mean?). If published, this will include your full peer review and any attached files.

Reviewer #1: No

Reviewer #2: No

---

## [Author Response · Author response to Decision Letter 0]

28 Sep 2021

We thank the editor and reviewer for this process and enabling us improve our study.

---

## [Decision Letter · Decision Letter 1]

5 Jan 2022

Spatial approach for diagnosis of yield-limiting nutrients in smallholder agroecosystem landscape using population-based farm survey data.

PONE-D-21-15324R1

Dear Dr. Muhati,

We’re pleased to inform you that your manuscript has been judged scientifically suitable for publication and will be formally accepted for publication once it meets all outstanding technical requirements.

Kind regards,

Remigio Paradelo Núñez

Academic Editor

PLOS ONE

Additional Editor Comments (optional):

Reviewers' comments:

Reviewer's Responses to Questions

**Comments to the Author**

1. If the authors have adequately addressed your comments raised in a previous round of review and you feel that this manuscript is now acceptable for publication, you may indicate that here to bypass the “Comments to the Author” section, enter your conflict of interest statement in the “Confidential to Editor” section, and submit your "Accept" recommendation.

Reviewer #1: All comments have been addressed

Reviewer #3: All comments have been addressed

2. Is the manuscript technically sound, and do the data support the conclusions?

Reviewer #1: Yes

Reviewer #3: Yes

3. Has the statistical analysis been performed appropriately and rigorously? 

Reviewer #1: Yes

Reviewer #3: Yes

4. Have the authors made all data underlying the findings in their manuscript fully available?

Reviewer #1: Yes

Reviewer #3: Yes

5. Is the manuscript presented in an intelligible fashion and written in standard English?

Reviewer #1: Yes

Reviewer #3: Yes

6. Review Comments to the Author

Reviewer #1: The paper has been improved through thorough revision. Majority of the responses given previously as to why some nutrients were limiting have been addressed.

Reviewer #3: The authors have improved their manuscript by considering the most important suggestions by the reviewers.

7. PLOS authors have the option to publish the peer review history of their article (what does this mean?). If published, this will include your full peer review and any attached files.

Reviewer #1: No

Reviewer #3: No

---

## [Editor Report · Acceptance letter]

24 Jan 2022

PONE-D-21-15324R1 

Spatial approach for diagnosis of yield-limiting nutrients in smallholder agroecosystem landscape using population-based farm survey data.

Dear Dr. Ichami:

I'm pleased to inform you that your manuscript has been deemed suitable for publication in PLOS ONE. Congratulations! Your manuscript is now with our production department. 

Kind regards, 

on behalf of

Dr. Remigio Paradelo Núñez 

Academic Editor

PLOS ONE